# Evolution of Early Onset Scoliosis under Treatment with a 3D-Brace Concept

**DOI:** 10.3390/jcm11051186

**Published:** 2022-02-23

**Authors:** Rebecca Sauvagnac, Manuel Rigo

**Affiliations:** Rigo Quera Salvá S.L.P., 08021 Barcelona, Spain; rebecca.sauvagnac@gmail.com

**Keywords:** early onset scoliosis, bracing, non-operative treatment

## Abstract

The objective of this study is to examine the evolution of all the braced patients diagnosed with early onset scoliosis in a private scoliosis center. All patients diagnosed with EOS and braced before the age of ten were retrospectively reviewed. The results have been defined in accordance with the Scoliosis Research Society (SRS) for bracing criteria, and with a minimum follow-up of one year. Improvement and stabilization were considered successful treatments, while failure was considered to be an unsuccessful treatment. Successful results were observed in 80% of patients (63% worst case). In the success group, the Cobb angle was reduced from 36.3° (21–68) to 25° (10–43), with 36% of patients being initially treated only with night-time bracing. Twenty percent of the patients failed, seven had more than 45° at the last control and five had undergone surgery. This study suggests that bracing, using a modern 3D-brace concept, could be an effective treatment option for early onset scoliosis and advocates exploring its effectiveness as an alternative to casting throughout studies of higher levels of evidence.

## 1. Introduction

Morphological scoliosis is a complex three-dimensional deformity of the trunk and spine that has multiple causes, which can appear during any period of life. When scoliosis has an unknown cause, it is called Idiopathic Scoliosis (IS). IS is the most common form of morphological scoliosis and appears in apparently healthy children during any growth period. The Scoliosis Research Society, in its revised glossary of terms [1], has established the chronological presentations of IS as: (1) infantile scoliosis: presenting from birth through age 2 + 11; (2) juvenile scoliosis: presenting from age 3 through age 9 + 11; (3) adolescent scoliosis: presenting from age 10 through age 17 + 11; (4) adult scoliosis: presenting from age 18 and beyond. Another common term is Early Onset Scoliosis (EOS). This term was used by Ponseti and Friedman to confirm a worse prognosis in scoliosis beginning before the age of 10 in comparison with scoliosis developing from aged 10 years and beyond, known as Late Onset Scoliosis (LOS) [2]. The term EOS was also used by Dickson as coinciding with infantile scoliosis [3], and later to define scoliosis present in children younger than 5 years of age [4]. Following some years of debate, the term EOS is now used globally and is defined by the SRS as a curvature of the spine ≥10 degrees in the frontal plane with onset before 10 years of age, including congenital, neuromuscular, syndromic and idiopathic. Thus, according to the SRS, EOS encompasses the two classical types: infantile and juvenile. In this paper, we will use this most-recently defined term for EOS, thus including infantile and juvenile.

EOS is a main priority for specialists due to its potential for early progression and, mainly in thoracic scoliosis, for its high risk of respiratory impairment. Currently, there are no treatment recommendations for these young patients. In 2007, Lenke [5] carried out a literature review and proposed brace treatment for scoliosis between 25° and 60°. The most widely used braces were the Milwaukee Brace (MB) and Thoraco-Lumbo-Sacral Orthosis (TLSO) with eventual preliminary serial casting. Brace treatment should be abandoned in favor of surgical treatment by spinal fusion or with growing-rods with or without preliminary halo-gravity traction in curvatures of more than 60°. Early surgery can risk lessened spinal height, chest-wall and lung growth [6] and the crankshaft phenomenon [7]. 

Even if new surgery with growth-friendly instrumentation limits these complications, they often suppose multiple surgeries with the associated risk of cerebral neurodevelopmental due to repeated anesthesia [8,9].

In 2011, the Pediatric Orthopaedic Society of North America (POSNA) published a survey on EOS patients from 195 practitioners [10], showing that 89.1% braced their patients (albeit with no mention being made of the type of brace used), 62% cast them, 64.1% operated on them employing growing-rods, and 39.1% used chest wall expansion. In 2016, Yang et al. [11] conducted a review of patients exhibiting curves of more than 25° and with more than 10° documented progression. Yang et al. proposed bracing to maintain correction obtained from serial casting in order to delay surgery. Otherwise, most surgeons consider a scoliosis progression over 60° to indicate the need for distraction-based implants, as well as spinal fusion at the end [12].

It is now well known that bracing is effective in preventing progression to surgery in adolescent idiopathic scoliosis [13], but no consensus exists for infantile and juvenile idiopathic scoliosis. The most typically recommended braces for these patients are still the classic Milwaukee (MB) and TLSO braces. Harshavardhana and Lostein [14], in a retrospective study from 1956 to 1999 in 125 patients treated by the MB or TLSO braces before 10 years of age, showed an overall success rate of 45%. Khoshbin et al. [15] described around 88 patients with JIS treated by the MB, TLSO or Charleston braces between 1982 and 2011 and showed that 28% of the curves were improved or stabilized, 72% were progressing, and there was a 50% surgery rate. In comparison with these relatively poor results, in 2014 De Kleuver et al. [16] reported a 100% success rate with the Cobb angle decreasing in 42% of patients following brace treatment. However, this was in a cohort of only seven patients, three of whom had initially been treated by cast. 

Although inspired by the old principles of correction from the ‘Casting Era’, some new braces do appear to be more effective in treating juvenile or infantile scoliosis. In 2014, Moreau et al. [17] studied the effect of a detorsion night-time brace pursuing Charleston principles, in 33 patients with brace onset at a mean age of 4 years and 2 months. They found a 67% success rate and a median Cobb angle reduction for success patients of 15° (3–27). More recently, Thometz et al. [18] have proposed an elongation-bending-derotation brace for infantile and juvenile scoliosis. This new brace system is made by using CAD/CAM in a corrective position and has been shown to achieve correction or stabilization in 75% of curves, with a progression incidence of only 25%.

The 3D brace used in this study is inspired by the so-called Chêneau brace and the French Casting Technique EDF. More than a type of brace identifiable with a specific curve pattern, it is a brace concept that can be recommended—using different techniques for design and construction—to treat most curve patterns in AIS and EOS. The principles of this brace have been described by its developer [19] and different authors have showed good results in treating AIS, but no data have been published on EOS [20,21,22]. We began using this brace concept for EOS in the late 1990s, introducing some modifications in the construction technique. 

The main objective of our study is to evaluate the effect that rigid bracing (3D TLSO) has in EOS according to SRS criteria of success. The secondary aims are to study this effect on an EOS sub-group of patients with a known graver prognosis (younger than 5 years) and to evaluate factors that can influence treatment success. 

## 2. Materials and Methods

Study design: This is a retrospective case series of all consecutive patients fulfilling inclusion criteria from 2007 to 2019. 

Inclusion criteria were as follows: (1) Cobb angle of 25° or more; or Cobb angle of 20° to 24°, with documented progression; (2) starting brace treatment before 10 years of age; (3) minimum follow up of 1 year. We do not recommend bracing in all the cases with a Cobb angle of 25° or more as a single criterion for treatment, but rather we combine physical examination, Cobb angle degree, angle of vertebral rotation, costo-vertebral-angle difference, and factors such as positive family history. However, although these factors are taken into consideration when deciding on bracing or not, we did not collect these data for all the patients in a systematic way, which could have been used to discuss prognosis factors or report as outcomes. When considering the potential indications for bracing (25–60°), we try not to break the rules and, in most of the cases, we did not. Six patients with a Cobb angle between 21° and 24° were braced, all of them with a documented progression after an initial period of observation. Independent of the Cobb angle, they also presented some sign or signs of poor prognosis (e.g., high axial rotation, CVAD > 25°, high ATI value). Three patients with a Cobb angle over 60° were recommended bracing. Taking the Cobb angle as the only indication value is not always in accordance with the best clinical practice. For instance, we have rejected bracing for patients with curvatures less than 60°, or even less than 50°, due to the very high degree of rotation and rigidity, amount of lordotization and angular rib shape, making them poor candidates for any type of TLSO. As such, they were all referred to an orthopedic surgeon. The three cases with a Cobb angle over 60° who did undergo bracing still had curvatures flexible enough to accept a TLSO, although they were also visited by the orthopedic surgeon. However, following an interdisciplinary team approach, we decided by consensus to continue with bracing, for several reasons in relation with individual characteristics to try and buy time. Some patients diagnosed with EOS, however, were simply observed with no specific intervention other than regular medical controls, and radiographic controls only when indicated from clinical observation and exploration. Thus, not all the patients with a Cobb angle of 25 degrees or higher were directly recommended for bracing. Nevertheless, the patients included in this present study are those treated with a brace who fulfilled the inclusion criteria defined above. Where possible (i.e., a better prognosis) and after discussing the case with the parents, an initial recommendation for wearing time was ‘night-time’ (i.e., 6 to 12 H). In the cases with poorer initial prognoses, these were prescribed ‘full time’ wearing (≥20 H) from the very beginning. That said, some parents would only accept ‘partial time’ (13–19 H), not full-time, for their children. We were not able to instigate in order to objectively report the real wearing time. 

Exclusion criteria were pre-treated patients (casting or brace).

At each visit, patients had their brace checked and underwent a physical examination. Because we have a private practice and, in accordance with an evidence-based personalized approach model [23], we do not have a fixed protocol for radiographic control, an Out Of Brace (OOB) radiograph was ordered when (and only when) either a physical examination gave us a clear suspicion of curve deterioration, or we had to change the brace due to growth and development, or we suspected a change in the curve pattern necessitating a new brace design. 

Before 2007, we had initiated treatment with the 3D brace concept in only four EOS cases without previous treatment (usually casting). Three of them were still under treatment when we started the informatics clinical files and data collection and were considered for inclusion in this study, albeit with initial data having to be added to the file retrospectively. The four patients were included, but one had incomplete data, and another was lost. Thus, this series includes all the EOS we have treated with a rigid TLS 3D brace, with no previous treatment; consequently, it is a series of consecutive treated cases.

The brace: The 3D brace according to Rigo principles can be either hand-made or built using CAD CAM. The basic idea is to design highly specific contact and expansion areas on a positive mould of the patient’s trunk. The contact areas are designed into the brace’s shape, orientation and level in order to produce the necessary combined detorsional forces. Thus, the brace follows, in most of cases, the simple general principle of correction defined from Dubousset (1992) [24], ‘reaching the best possible frontal and sagittal alignment by using detorsional forces’, and some well-described specific principles of correction: (1) regional derotation plus cranial and caudal counter-rotation forces; (2) lateral as well as ventro-dorsal contacts for frontal and sagittal plane alignment guidance; (3) a special mechanism to fight against the structural lordotization of the thoracic region, which depends on the level, the shape and the orientation of the contact areas of the brace. As a whole, the objective is not the maximum correction in one single plane but rather reaching the best possible frontal and sagittal alignment while preventing any worsening of compensatory curvatures and/or imbalance (Figure 1 and Figure 2). Thus, the level-orientation shape of where the detorsional forces are applied is of equal importance to the standard of the brace, as its implementation is in accordance with the clinical type and radiological curve pattern. The level of the main derotational force is the apical region of the main curve or curves, with counter-rotation forces acting at the proximal thoracic region and at the lumbo-pelvic or pelvic region. The design of the contact areas can then be made using CAD CAM or classical hand-made rectification of the pre-elongated mould. As a result, this is not a standardized orthopedic product but rather a treatment concept needing specific knowledge and experience from those prescribing, manufacturing and later controlling the brace. Once again, the knowledge and experience of the treatment team, as noted by the SOSORT guidelines for brace treatment [25], are essential areas of expertise.

Consistent with SRS criteria [26], we defined improvement to be a decrease of more than 5° of the Cobb angle between brace initiation and the final control, stabilization a Cobb angle variation ±5°, and failure an increase of more than 5°, Cobb > 45° at last control or at maturity, or a final need for surgery.

Improved and stabilized patients were all included in a simple success group. Thus, treatment could be defined as success or failure and patients as responders or not responders. 

Because of its higher potential of progression, we also separately analyzed patients that could have originally been defined as EOS according to Dickson, i.e., those with a relevant spinal deformity before 5 years old. 

Statistical analysis was performed using PSPP and GNU Software. 

For quantitative variables, we analyzed means, standard deviations and ranges. 

To evaluate the correlation between Cobb angles before and after treatment, we used a *t*-test for the paired sample, and to evaluate factors that could have influenced the success of the treatment we used a one-factor ANOVA. 

## 3. Results

From a list of 280 patients younger than 10 years old coming for a consultation because of a suspicion of scoliosis and seen for at least a second time, we confirmed 84 EOS patients according to the SRS definition. Sixty-six patients fulfilled the inclusion criteria. Nine out of the sixty-six were excluded due to previous treatment (brace or cast). Twelve out of the fifty-seven patients finally included had incomplete (10) or lost data (2). Forty-five patients with complete data were fully analyzed and a worst-case analysis was then performed in terms of ‘rate of success’ considering the 12 patients, with lost or incomplete data, as failures (Figure 3).

Thus, 45 patients were finally analyzed, with a female–male ratio of 2:1.

Most of the patients (77%) had idiopathic scoliosis. Ten patients had a known cause: one congenital, five neurological, one Beals syndrome, one arthrogryposis, one Ehler Danlos syndrome, and one neurofibromatosis syndrome. 

It was not possible using the Lenke classification to describe the radiological curve pattern of this present population. Using SRS terminology, however, 69% of the cases were single thoracic without lumbar, with functional lumbar or with a compensatory structural lumbar curve (Rigo clinical types A and C). Twenty-four percent of the cases were real double thoracic and lumbar or thoracolumbar (Rigo clinical type B), while only 4% were single lumbar or thoracolumbar (Rigo clinical type E). Figure 4 shows the distribution according to the Rigo classification [27].

The mean Cobb angle at brace initiation was 36.1 ± 11.6° (21.0–68.0). Age at first visit was 6.6 ± 2.0 years (2.5–9.8), age at last control was 13.2 ± 3.0 years (6.2–20.7). Mean years of follow-up was 6.5 ± 3.0 (1.0–15.6).

### 3.1. Treatment Success and Failure

Eighty percent of the patients met the success criteria at the last control: 49% improved and 31% stabilized their curves. Forty percent of all patients reached bone maturity (≥Risser 3 European Scale/≥4 American Scale) and 50% were still under a bracing regime at the last control. 

In the success group (improved or stabilized), we observed a significant Cobb angle improvement from 36.3 ± 11.4° (21–68) to 25.0 ± 8.6° (10–43) when comparing initial values at the beginning, before starting brace treatment, and values at the last control (*p* < 0.001 by *t*-test Student). In the group of patients showing improvement (22 patients), the Cobb angle decreased by 17.1° ± 9.1 (7.0–37.0).

In the success group (improved or stabilized), 53% were treated initially using a full-time brace, 36% with a night-time brace and the remaining 11% with a part-time brace (Figure 5).

Twenty percent (nine patients) failed, with a mean time of follow-up from the initiation of treatment to failure of 7.1 years ± 1.9 (5.2–10.1) and a mean progression of the Cobb angle of 22.7 ± 12.5° (10–46). Seven out of these nine patients had more than 45° at the last control (Figure 6). All the patients had been instructed to wear the brace full-time; however, we cannot report about the real compliance in these failed patients.

Treatment strategies for six out of the nine patients was changed when failure was detected: (1) increasing wearing time from night-time to full-time in four patients; and (2) two patients were recommended to undergo surgery. Three more patients had surgery sometime later, so, overall, five patients were finally operated on. 

### 3.2. Prognosis Factors 

When comparing the two groups showing success or failure, we did not find any significant difference in age (6.5 vs. 5.4 *p* 0.175), Cobb angle (36.3° vs. 35.6° *p* 0.875) at the initiation of brace treatment, and age at the last control (13.1 vs. 11.8 *p* 0.776) (Table 1).

### 3.3. Population under 5 Years of Age

Fifteen of the 47 patients were less than 5 years of age when they were diagnosed with EOS. As they are expected to have a poorer prognosis, we decided to analyze them separately. As expected, the female–male sex ratio was more balanced: 0.88:1 in this sub-group (vs. 2.13:1 in our total cohort).

Etiology was idiopathic scoliosis (73%), while others were congenital (1), neurological (2), and Beals syndrome (1).

Radiological curve patterns were mostly thoracic curves (67%) (A and C Rigo clinical type). Real double was observed in 27% of the cases (B Rigo clinical type). There were no single lumbar or thoracolumbar curves (Figure 2).

Follow-up time was not significantly different in this sub-group when compared with the whole cohort (Table 2). 

Only six out of these 15 EOS patients started brace treatment before 5 years of age. Mean age at brace onset was 4.6 ± 2.3 years old (1.0–9.0).

Sixty-seven percent of the EOS patients from this sub-group showed treatment success. The 27.8° ± 7.4 (15–40) Cobb angle at the last brace control was significantly lower compared with the Cobb angle at the initiation of brace treatment 35.8° ± 10.0° (29–63) (*p* = 0.027, by *t*-test Student).

As in the total sample, age and Cobb angle at the initiation of brace treatment was not different in both groups showing success or failure in this sub-group (Table 3).

## 4. Discussion

**Comparison with previous studies.** This is a cohort of 57 consecutive patients diagnosed with EOS (mostly Idiopathic) treated with a 3D-brace concept according to Rigo principles before the age of 10. Twelve patients were lost or presented incomplete data and were not included in the final analysis. We report an 80% success rate, defining success according to the SRS criteria of improvement or stabilization. In a worst-case analysis, the success rate was 63.2%. Our results are along the same lines as those reported by Moreau et al. [18] and Thometz et al. [19], both using new-generation 3D braces. Moreau et al. [18] had 67% of success with a mean Cobb angle reduction of 15°, a little bit lower than ours, but with all their patients receiving night-time bracing and being a little younger than ours. Notwithstanding this, a relevant proportion of our patients were instructed to wear their braces only at night, at least initially, and we did not find the age of the initiation of bracing to be different in our successful and unsuccessful patients. Moreau et al. used a detorsional night-time brace, only wearable at night due to its design. According to the description of the authors, the design of this brace was inspired by the Charleston bending brace and can be used only at night, but the horizontal plane action follows the Chêneau principles like our 3D brace. However, our brace does not have a different design when it is to be used only at night or to be worn full-time. We use a unique day/night design based on detorsional forces, which can be worn indistinctly at night or full-time, thus allowing us, when necessary and with greater flexibility, to increase the wearing time with no need to change the brace, yet still working well. Our results must be taken with precaution because, although 60% of our patients were scored Risser 3 (European), so able to be considered mature enough to stop bracing, our follow-up (6.5 years) was lower than the one reported by Moreau (10 y). Thometz et al. [19] reported around 75% of success (improvement or stabilization) in treating IS and JS with an Elongation Bending Derotation Brace. However, this report studies 38 patients followed only during their first year of treatment and must be considered as preliminary results.

What is important to mention is that, with all the limitations discussed below, our study, as well as the two previously discussed studies, suggests that new brace designs are superior to the classic Milwaukee and TLSO braces, as reported by Harshavardhana and Lonstein [15] and Khoshbin et al. [16], the two main papers about bracing in infantile and juvenile scoliosis. Harshavardhana and Lonstein report an overall success rate of 45%, defining success as ‘not reaching a surgical value’. It is true that the sample size of these authors is greater (*N* = 125 patients); however, this sample comes from a long-term database and represents only 15% of the juvenile cases. From a database of 841 patients, 80% were excluded for different reasons and, although some of these patients were observed, many were undoubtedly treated with a brace. Our smaller series represents all the patients we have treated with this type of brace (some are still under treatment). The pre-brace age is, on the other hand, higher in the study from Harshavardhana and Lonstein [15] (8 years) in comparison with ours (6.6), while the initial mean Cobb angle is lower (30° vs. 36°). Moreover, 32 out of 125 patients in their study initiated brace treatment during adolescence. According to Donzelli et al. [28], patients diagnosed during the juvenile period but braced later, during adolescence, can be considered, in terms of prognosis, as adolescent scoliosis. It is true that, like us, they did not find age differences in successful and unsuccessful patients, but rather a higher pre-brace Cobb angle was observed in those failing. Thus, we can consider our population of having a poorer prognosis in comparison to those from Harshavardhana and Lonstein [15], as we did not include any patient braced at or after 10 years of age, and our patients also have a higher pre-brace Cobb angle. In contraposition, follow-up is longer in their study and some of our patients are still under treatment and have not yet reached their adolescent growth spurt. Patients needing surgery in the study of Harshavardhana and Lonstein [15] were stable during the juvenile period and did show progression during the adolescent growth spurt, so we could expect that some of our patients will fail and might finally need surgery. Notwithstanding this, the mean chronological age in our series, as well as the bone age (60% at Risser 3-European), indicates that a relevant number of patients already passed the peak of growth and are not far from becoming mature. On the other hand, reaching that period of growth with a successful result is remarkable, as one of the objectives of non-operative treatment in EOS is delaying early surgery, in other words, buying time. Thus, we must be prudent in affirming that our results, in correlation with those from Moreau et al. and Thometz et al., suggest that a new-generation brace might be more effective in preventing progression in EOS in comparison with the Milwaukee brace or the classic TLSO.

We cannot compare our study with the study from Khoshbin et al. [16]. In this last study, in a cohort of 88 patients diagnosed with JIS, the authors showed a 50% incidence of final surgery, with 28% of patients showing a minimal curve progression or improvement. On the other hand, they reported a low compliance (49% wore the brace full-time). Although this study is considered one of the main references in relationship with the effect of bracing on juvenile idiopathic scoliosis, the mean age at diagnosis was 8.4 years and the mean age at the initiation of bracing was 9.3 ± 1.5 years for the whole sample and 9.9 ± 1.4 for those not needing surgery. This clearly indicates that a highly relevant number of patients started wearing the brace at 10 years of age or older, coinciding with the ascending phase of growth in the adolescent period. The Cobb angle at baseline was 31° (20–71). According to Donzelli et al. [28], the response to bracing in patients diagnosed with JIS according to the SRS (from 3 years to 9 years and 11 months) but starting bracing at 10 years of age or later is not different to those diagnosed with AIS. Thus, we could expect a response similar to that of children with very early AIS in the Khoshbin et al. study (initiation of bracing and pre-brace curve magnitude), and, knowing that bracing is dose dependent in AIS [14], the high incidence of final surgery should be no surprise, considering a compliance under 50%. 

**Prognosis factors.** In relationship with prognosis factors, we did not find any factor that could explain success or failure as a result of the treatment. Indeed, neither age at brace initiation, pre-brace cobb angle, nor initial wearing time are significantly different between those responding well and those failing. In the Moreau study [18], the unsuccessful group was older (58 months vs. 42 months) and had a higher main Cobb angle (35° vs. 28°) at the initial examination, compared with the success group. Our results could indicate that pre-brace age and Cobb angle are, contrary to the study from Moreau et al., not factors for a poorer treatment response, while having the chance to change from nigh- time to partial or full-time when necessary are. However, we must admit that with age and curve magnitude both being factors for prognosis when looking at the natural history of EOS, our results could be different with a bigger cohort and the exclusion of non-idiopathic cases. On the other hand, we did not look at compliance and in-brace correction as possible factors associated to brace response. We cannot report on real compliance as we did not use sensors. In any case, we have the impression that, once having discussed the proper wearing time with parents and accepting their final decision about this being full-time or partial night-time, compliance in EOS patients is good, as it depends mainly on parental care, at least before and during the ascending phase of growth in the adolescence or before menarche. We had very few cases where compliance was good during the pre-menarche period and failed afterwards. It looks to be easier for children wearing a brace before puberty to continue using the brace with good compliance during the more difficult time of adolescence in comparison with those starting bracing when they are already adolescents; this is, however, just a subjective impression. In-brace correction was not analyzed because, in accordance with our clinical protocols, we only look at in-brace correction in those patients that are recommended to wear the brace full-time, as in-brace correction as a prognosis factor is highly dependent on the wearing time [14]. Consequently, in most of our EOS patients, radiological controls are made out-of-brace. 

**The sub-group of patients younger than 5.** Regarding the sub-group of EOS patients younger than 5 years of age with a relevant structural curvature at presentation, the 15 patients had the same characteristics as the whole cohort. As is known in the literature, we found a female–male ratio that tends to be towards boys (0.8:1) and a higher number of thoracic curves [29,30], and this could be interpreted as these patients being part of the real infantile scoliosis group, those going into progression during infantile and childhood periods before the properly defined juvenile period [31]. As in the entire cohort, we did not find any possible factors associated with success or unsuccessful results. It is noticeable that only six of this sub-group of 15 EOS patients diagnosed before the age of 5 years, were really braced at or before the age of 5 years. Two of these six patients failed and four succeeded (*p* = 0.67 by chi-2). However, the type of study and the number of cases analyzed in this particular sub-group do not allow us to draw any clear conclusion. The proportion of failures in this sub-group (5/15) in comparison with the whole sample (9/45) strongly suggest that this sub-group of patients has a poorer prognosis; consequently, we still believe it would of interest to analyze this sub-group of patients and those diagnosed at six years of age to 9 years and 11 months separately, as most probably there are two different conditions.

**Good responders and night-time bracing.** Looking at this present study, it is important to point out the apparently good response to night-time bracing. Early onset scoliosis with a Cobb angle over 25° is generally considered as having a poor prognosis. However, the Cobb angle by itself was found not to be the main factor associated with the risk of progression in EOS, but rather in combination with other more important factors, such as the amount of axial rotation [32] or the costovertebral angle difference [33]. Furthermore, independent of the initial Cobb angle, flexibility and the shape and magnitude of the rib hump could be determinant factors for a good response to treatment [34]. In our current clinical practice, we do not use a closed protocol based on a particular Cobb angle to initiate bracing, but we make a decision (in consensus with the family) based on a personalized approach and by taking into consideration values such as axial rotation, costovertebral angle difference, flexibility (tested only by clinical exploration) and the shape of the rib hump. We consider these factors to be important not only in brace indication but also in the prescribed wearing time. In AIS, there is consensus about recommending a wearing time of ≥18 h in scoliosis involving a high risk of surgery [34]. It is true that EOS is a population classically considered to be at high risk of needing surgery but, still, some patients with the diagnosis of EOS according to the SRS definition do not go into progression before the age of 10 but do so later during the adolescent period, showing the same behavior as classical AIS [28], while others do not progress or even regress. Thus, there was some data based on clinical experience and evidence supporting the expectation of good responders that might be treated with night-time bracing, and these results support the prescription of night-time bracing in some selected patients, i.e., basically those with relative degrees of rotation (<10° Perdriolle), Cobb angles under 35°, low costovertebral angle differences (<20°), with flexible curvatures and ‘round versus angular rib humps’. 

**Limitations.** Our study has important limitations. First, it comes from a unique center, showing results from a brace made by the main author of this article. Even if the main outcome was evaluated from a universal and reproducible measurement like the Cobb angle is, it would be necessary to investigate the effectiveness of this type of 3D brace when used by others. Notwithstanding this, the fact that our results are akin to those using similar brace principles minimizes this weak point. Second, a more important limitation is that this is a retrospective study with no control group, so conclusions will need to be in accordance with this low level of evidence. However, considering the very few studies of these characteristics and the general belief that casting is the main treatment option in EOS, we felt there is a strong need to report our experience to increase the amount of evidence in order to justify a prospective study with the highest possible methodology, comparing casting and bracing in EOS. After looking at our results and some parts of the previous discussion, we would suggest separately analyzing those EOS developing a relevant structural scoliosis before the age of 5, defining well what means ‘relevant’, with the need of bracing before the age of 5, rather than just taking the classic definitions of IIS and JIS. Idiopathic scoliosis and non-idiopathic should also be analyzed separately. Furthermore, it would be interesting to explore the possibility of ‘serial bracing’ in the youngest patients (i.e., younger than 3 years). This is something that we have already tried with a few patients when families rejected casting for various reasons, and after being informed about current evidence. The CAD CAM technology makes this easier, with no need to stress the child unnecessarily. Once the design has been made, the real cost of fabricating the plastic itself is not so high and, while it is necessary just to enlarge and elongate the brace for ‘serial bracing’, in general, it is not necessary to change the design significantly. 

In retrospective studies, patients lost for the final analysis are always a problem. We had to exclude two lost patients and ten with incomplete data. Thus, we made a worst-case analysis (see Figure 3) by adding to the forty-five analyzed patients the ten with incomplete files and the two lost patients but, even so, we still obtained a final success percentage of 63.2% (36/57), with a percentage of unsuccessful cases of 36.2% (21/57). These numbers are still interesting, as they are closer to everyday clinical practice.

A further limitation is the fact that this series includes non-idiopathic scoliosis. First of all, we wanted to report on our experience in treating EOS with bracing, and the definition of EOS includes types of scoliosis other than idiopathic. On the other hand, the incidence of neurological anomalies in EOS was still a controversial issue at the time we started bracing in EOS. We were not routinely ordering MRIs at that time, but just in case we found some specific atypical features. In 2004, Morcuende et al. reported a relatively low probability of neurogenic lesions in EOS with atypical features other than severe curves or neurological abnormalities (3% probability) [35]. Thus, we ordered MRIs only in the cases where we found a severe curve or neurologic changes. We found three patients with an anomaly in the neuro axis (Chiari and Syrinx). Two of them had been operated on. One boy who had had surgery before the age of 3 did not show a regression of the curvature and he was consequently recommended to undergo bracing. The second patient was braced at the end of 7 years after observing a progression from 44° to 55° after neurosurgical decompression. The third patient was treated directly with bracing with no previous neurosurgical intervention, and she showed a spontaneous resolution of the syrinx. The latter two patients mentioned were presented in a double case report during the SOSORT meeting of 2012 in Milano [36]. All three patients showed a good response to bracing and did fall into the success group. 

Other important outcomes that we were not able to report on in detail are breathing function, trunk shape, sagittal radiological values and quality of life. The first two, breathing function and trunk shape, especially the rib cage, are partially related. We believe that allowing the correct development of the rib cage and re-shaping the rib cage would be important factors for future breathing function and curve stability. It must be pointed out that, historically, many scoliosis specialists have advised against using TLSO in EOS due to the risk of worsening breath function and deforming the ribs. We must recognize that the type of brace used in this study, due to its sophisticated design, has an iatrogenic potential when used improperly and only looking at the Cobb angle correction. Sagittal radiological values are, nowadays, considered to be highly important from both clinical and scientific points of view. However, when these young patients started their treatment, the consensus about using initial radiographs in the lateral projection was not so clear, and we tried to use the fewest possible number of radiographs in this young population. Quality of life, from a biopsychosocial perspective, is another important aspect to consider. Although we routinely assess quality of life in adolescents and adults using validated instruments, we did not look at this data in this present study. We have the intention to look at all these important aspects (breathing function, trunk shape, sagittal alignment and balance and quality of life) in this same reported population once they have all finished their treatments with a minimum follow-up of two years. That said, we felt this was a good time to report on our experience with non-operative treatment in EOS.

## 5. Conclusions

This study suggests that bracing, using a modern 3D-brace concept, could be an effective treatment option for EOS and advocates exploring the effectiveness of bracing in EOS through studies of a higher level of evidence. 

## Figures and Tables

**Figure 1 jcm-11-01186-f001:**
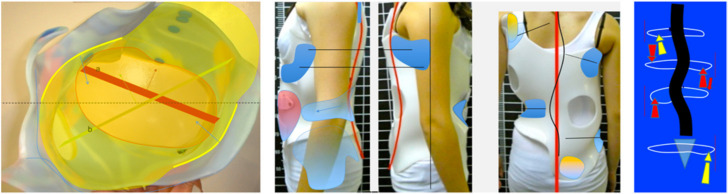
The brace design when treating a double structural scoliosis (Rigo B Type). Two mechanisms for regional derotation are applied, one at the main thoracic region and another at the lumbar/thoracolumbar region (red arrows). These two mechanisms for regional derotation work in combination with two counter-rotation forces (yellow arrows). Contact areas are provided, laterally, ventrally and dorsally, to guide the frontal as well as the sagittal plane alignment and balance. (The figures describe the brace design showing an adolescent. The brace design in EOS follows the same principles).

**Figure 2 jcm-11-01186-f002:**
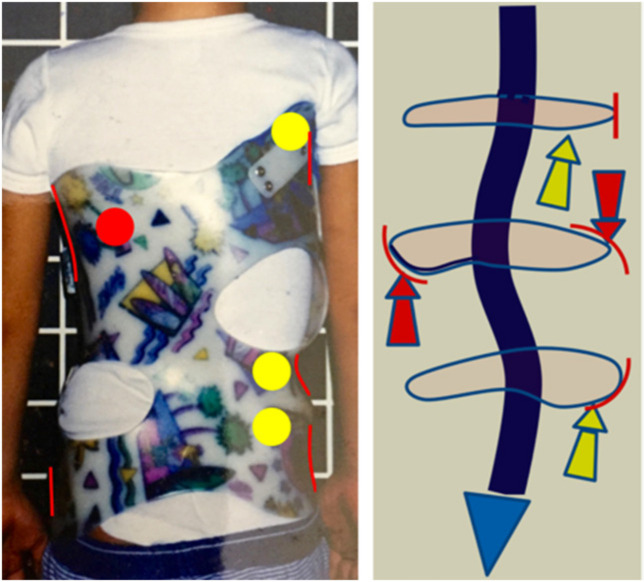
The brace design in a single thoracic scoliosis (Left convex in this EOS patient). One single mechanism of regional derotation (red arrows) works in combination with two counter-rotation forces (yellow arrows). Lateral, dorsal and ventral contacts are provided to guide frontal and sagittal plane alignment and balance.

**Figure 3 jcm-11-01186-f003:**
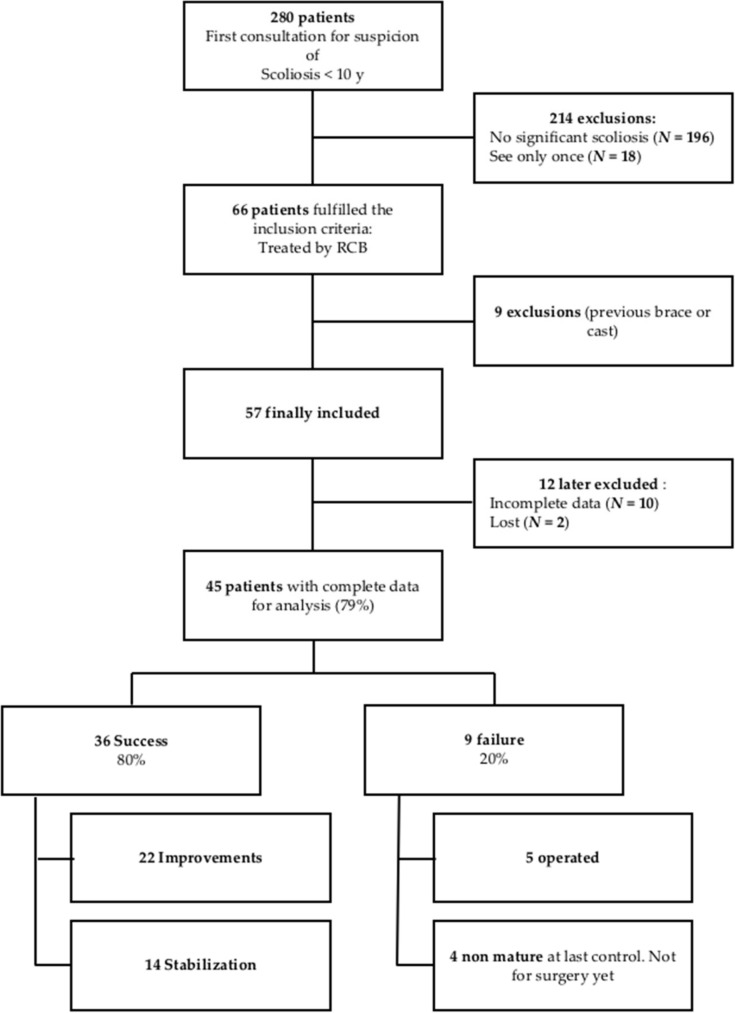
Flow chart. Worst case analysis = 63.2% success (36 success/57 finally included, presuming that cases lost to follow had failed).

**Figure 4 jcm-11-01186-f004:**
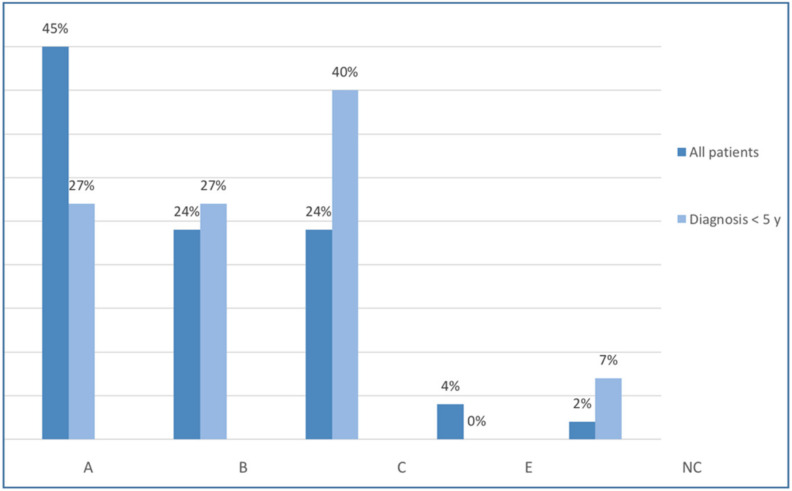
Distribution of clinical types according to Rigo Classification (A: Rigo type A; B: Rigo type B; C: Rigo Type C; E: Rigo Type E; NC: Not Classifiable).

**Figure 5 jcm-11-01186-f005:**
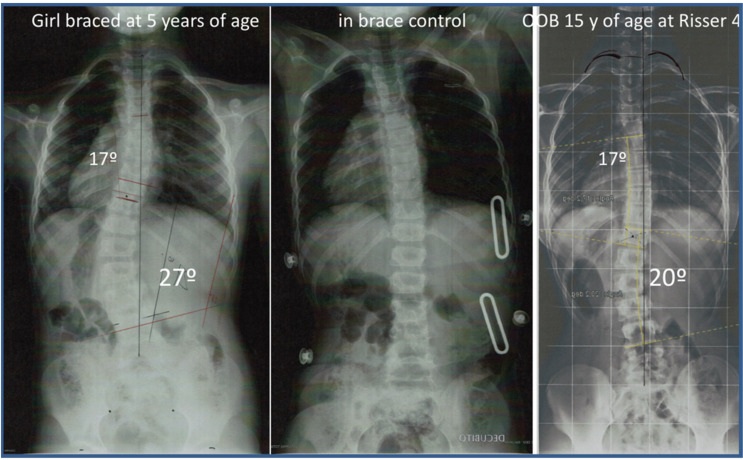
Example of success. Result of treatment in a girl braced at 5 years of age with a left lumbar scoliosis of 27°, completing treatment under night-time regimen until 15 years of age at Risser 4, stopping treatment with a left lumbar curvature of 20°.

**Figure 6 jcm-11-01186-f006:**
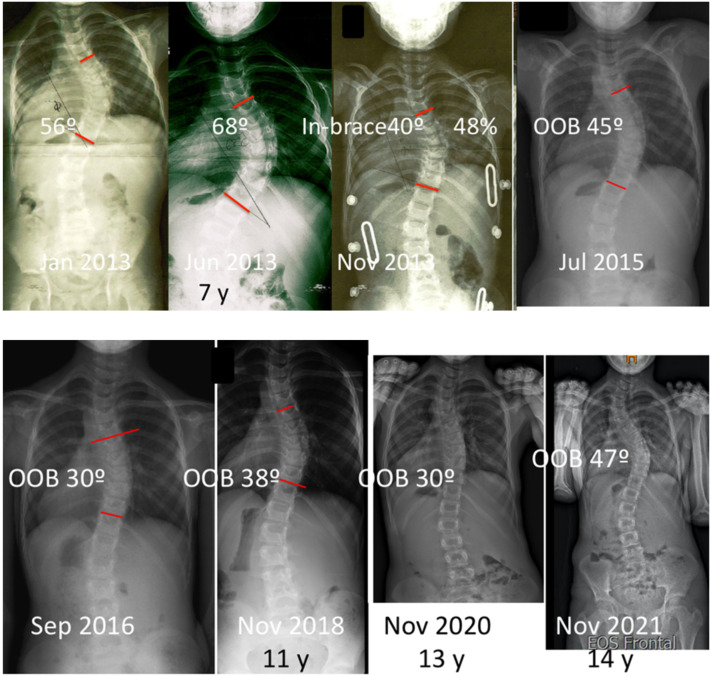
Girl treated with full-time brace, initiating the treatment at 7 years of age with a 68° thoracic scoliosis. With a 48% in-brace correction in her first brace she showed a good response until 10 years with a scoliosis of 30°. She was followed until 12 years of age still with a scoliosis of 30°. Next control at 13 years, she was still stable with 30°, but had developed a more relevant proximal curvature (after closing this present study). Between 2020 and 2021 (not registered in this present study), she showed a deterioration, with the development of a proximal curve of 43°, forcing us to stop bracing and recommending her to undergo surgery.

**Table 1 jcm-11-01186-t001:** Cobb evolution in both successful and failure groups (significance *p* < 0.05).

	Successful Group (N36)		Failure Group (N9)		
	Mean	Range	Mean	Range	*p*
Age at brace onset (years)	6.5 ± 2.0	2.0–9.0	5.4 ± 2.5	1.0–8.0	0.175
Cobb angle at brace onset (°)	33.3 ± 11.4	21.0–68.0	35.6 ± 13.1	24.0–65.0	0.875
Age at last control (years)	13.1 ± 3.2	6.2–20.7	12.8 ± 2.7	7.0–16.0	0.776
Cobb angle at last control (°)	25.0 ± 8.9	10.0–43.0	58.2 ± 14.1	38.0–76.0	<0.001

**Table 2 jcm-11-01186-t002:** Baseline characteristics of study population (significance *p* < 0.05).

	All Patients	Diagnosis < 5 y	*p*
No. of patients	45	15	
Age at diagnosis, Mean ± SD (y)	5.3 ± 2.3	2.7 ± 1.2	<0.001
Sex			
Female	67% (30/45)	47% (7/15)	
Male	33% (15/45)	53% (8/15)	
Age at brace onset, Mean ± SD (y)	6.3 ± 2.1	4.6 ± 2.4	0.012
Age at last control, Mean ± SD (y)	13.1 ± 3.1	11.6 ± 2.9	0.106
Years of follow up, Mean ± SD (y)	6.5 ± 3.0	6.6 ± 2.3	0.865

**Table 3 jcm-11-01186-t003:** Patients under 5 years at diagnosis. Cobb evolution in both successful and failure groups (*p* < 0.05).

	Successful Group (N10)		Failure Group (N5)		
	Mean	Range	Mean	Range	*p*
Age at brace onset (years)	4.8 ± 2.6	2.0–9.0	4.2 ± 2.8	1.0–8.0	0.667
Cobb angle at brace onset (°)	35.8 ± 10.0	29.0–63,0	40.4 ± 16.6	24.0–65.0	0.511
Age at last control (years)	11.1 ± 3.3	6.2–16.7	12.7 ± 1.6	10.9–14.8	0.378
Cobb angle at last control (°)	27.8 ± 7.4	15.0–40.0	66.2 ± 13.2	43.0–76.0	<0.001
Cobb evolution (°)	8.0 ± 9.6	−3.0–25.0	−25.8 ± 15.5	−46.0–−11.0	<0.001

## Data Availability

The data presented in this study are available on request from the corresponding authors. The data are not publicly available due to privacy.

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
