# Peer review of "Evolution of Early Onset Scoliosis under Treatment with a 3D-Brace Concept"

_jcm, 2022, doi:10.3390/jcm11051186_

Round 1
Reviewer 1 Report
The Introduction can be simplified. Why not chose one accepted classification, eg the SRS classification. Present previous data for EOS diagnosis and 3D bracing.
In materials it is important to present the selection of patients. Why is only 66 out of 200 patients included. Are patients included consecutively or chosen from the practice?
Lost patients are not an exclusion criteria.
The follow up is short, patients have not passed growth spurth spurt.
Author Response
The authors thank the reviewer for the concise comments and suggestions, which, we will try to answer and change in order to improve the quality of the paper.
Comment and suggestion number 1: we understand this and have tried to reduce the introduction section, but we must admit we have been very successful. We still feel it is important to describe the different accepted terminology for the EOS population. The SRS classification of Idiopathic scoliosis is covering Idiopathic EOS but EOS is more than idiopathic and the accepted by the SRS definition of EOS is just referred to the presence of a curvature measuring 10 degrees or more in a patient younger than 10, no matter the cause is congenital, syndromic, neuromuscular or idiopathic. However, this definition of EOS says nothing about the risk of progression and so many patients defined nowadays as EOS will show progression later during the adolescence, or will never progress or even will regress. We think it is important to remark these facts in the introduction section. We have also added something in relationship with this issue in the discussion section.
Comment and suggestion number 2. We have describe better the way the patients came into this study. We are sorry that this was not well described in the submitted version. We found 280 patients under 10 years of age consulting because the suspicion of scoliosis and coming at least for a second consultation. We found 88 patients out of 280 with the final diagnosis of EOS. Some of these 88 patients were and have been just observed showing no a progressive curve to be treated with a brace and some were directly referred to the orthopedic surgeon because we found they were not good candidate for bracing. This has been changed in the material and method section and can be also see in the flow chart (results section)
Comment and suggestion number 3: We agree, absolutely, and we cannot justify this unexplainable methodological mistake. This has been changed in the material and method section and so in the results section. Flow chart shows this change. We just did not include the 12 patients with incomplete data or lost into the final analysis, but we made originally a worst case analysis with this not really excluded patients and did report about that. We hope that the way we present it now is more acceptable from the methodological point of view.
Comment and suggestion number 4. We think this is properly addressed in the discussion section. The number of patients that have passed already the peak of growth is relevant and sure, some patients will still show progression and failure (like did happen in the case from figure 4, who showed progression after the study was closed). But we still think that bringing these patients with a bad prognosis, with such proportion of patients being already in the deceleration phase (Risser 3) into this situation is something remarkable. We agree, this is not enough to drawn any final conclusion about real success rate, but delaying surgery is also a good objective in this population, and we think these results show the realistic expectation of at least delaying surgery. We are very cautious in the discussion and conclusions. We do not say anything in the conclusion about this paper demonstrating the effectiveness of bracing in EOS. We just suggest that 3D bracing could be an effective treatment in EOS, at least more effective than the old Milwaukee brace and the classical Boston (not new Boston versions that are also following 3D principles), in a way that this should be investigated with the highest possible level of evidence. At the present state of the art we believe these paper, in spite of its low level of evidence, has quality enough to be published.
Once again, we thanks the reviewer for not rejecting directly the paper, something that we would understand due to its low level of evidence, and we appreciate very much his/her comments and suggestions that we have tried to follow with the highest interest.
Rebecca Sauvagnac and Manuel Rigo
Reviewer 2 Report
This is an interesting paper on a topic that is in need of further research and high-quality evidence. The authors have done a good job presenting this manuscript. I have some remarks and concerns listed below which can hopefully be clarified.
1) Generally, EOS patients have a higher rate of neuroanatomical abnormalities; such cases usually require MRI to exclude such pathology. How many of these cases were investigated with MRI and what was the rate of findings? Some individuals were not idiopathic as presented in the manuscript, but for those labelled idiopathic; were they fully investigated?
2) In terms of compliance; did the authors make an attempt to monitor these with heat sensors?
3) The authors describe that patients received number of hours in brace recommended based on expectations on worse outcome. What where the criteria assessed? In my opinion, all patients below the age of 10 should be considered high risk when having a curvature surpassing 25 degrees. Further, what is the rationale with nighttime bracing for such a high risk population?
4) Did the authors have any sagittal parameters assessed? Sagittal radiographs are of importance when managing idiopathic scoliosis clinically but also from a scientific point of view very interesting.
5) There were 15 patients below the age of 5 and some also below the age of 3. In my opinion, bracing is very difficult in these ages. What conclusions to the authors draw on this topic based on the findings? Additionally, the mean age at last follow-up for this cohort was 11.6 years, one can expect that they had still significant remaining growth. Is there data on their status at skeletal maturity? Furthermore, can one really say some are successfully treated when last follow-up is at 11 years of age?
6) Were there any attempts from the authors to gather patient-reported outcome measures from patients and parents? I understand that these are young children but such data is always interesting to present. Was there at least an attempt at later follow-ups when the children were older?
7) There seems to be patients having curves surpassing 50 degrees and the largest curve measures 68 degrees. What was the rationale for using a brace on these individuals? Why was not surgery proposed instead?
Author Response
We appreciate very much the comments made by the reviewer and will try to do our best in giving an answer to his/her concerns.
1) We agree. At the time we started this series, we were only ordering MRI in patients with severe curves and/or with neurologic changes, in accordance to Morcuende, Dolan, et al findings. We have added a complete paragraph in the discussion about this issue. We made a mistake indeed. We reviewed the files and found three patients that went initially to the idiopathic group while presenting some abnormalities in the neural axis (Chiari I and syrinx). We considered them first like presenting an idiopathic like pattern in association with these neurologic abnormalities. These 3 patients are the positive cases presenting neurologic abnormalities, out of 11 patients that were explored with MRI. In our private clinical practice we have been using restrictive criteria to order MRI (severe curve over 45º and/or neurologic changes). Unfortunately we cannot confirm that the rest of the patients are all real idiopathic cases. Although the 3 patients had idiopathic like pattern rather than typical neurogenic curve pattern, we have separated them from the idiopathic group and have summed to the neurologic group. The three patients did show a good response and two of them were presented as a double case report at the SOSORT meeting in Milan (2012), when the treatment was not finished yet. In this present series the three patient finished their treatment and showed a good response. One of the girls started her treatment with a Cobb angle of 55º after a documented progression from 44º , which occurred after neurosurgical decompression. This patients are reported to respond very bad to bracing. This girl responded well and reached full maturity with a residual thoracic curvature under 20º.
2) We are sorry. We do not use sensors in our braces so we cannot report about real compliance. For ethical reasons, our private clinical practice does not allow us to use treatment strategies that are no so relevant for the efficacy of the treatment and might raise the price of the treatment with no need. It could be argued that sensors might also increase compliance but in the private clinical practice and EOS, compliance is practically guarantied by highly motivated parents. We understand about the scientific relevance of this point and we are aware that we would need to use them in case of participating in a research of the highest level of evidence, but unfortunately we do not use them routinely and so we cannot report about it as explained now in the discussion section.
3) We think this is now better described in the material and method sections as well as in the discussion section. We believe that the Cobb angle alone is not the best indicator of the risk for progression in EOS and so we do not use a simple protocol based on the value of the Cobb angle alone. We use to combine different parameters in combination with the age and the Cobb angle, like axial rotation (or better to say torsional angle in accordance to Perdriolle), the Costo-vertebral angle from Mehta, the flexibility (clinical assessment), the shape of the rib hump ('round versus angular') and others to make a final decision about the indication for bracing as well as for the prescribed wearing time. We hope we have explained this with sufficient detail in the two above mentioned sections.
4) We fully agree. We currently use lateral radiographs to assess spino-pelvic alignment and balance according to the new evidence. Sagittal alignment and balance is of the highest importance in adult spinal deformities (AsD) and it is gaining attention in adolescent population as it is shown in the last paper of the SRS establishing consensus on the best practice guideline for the use of bracing in AIS (Roye et al Spine Deformity 2020). We have not so many data about the relevance of the sagittal alignment and balance in EOS and most of the orthopedic surgeons in our place use to order lateral radiographs only when they are planning surgery, but not to monitor non surgical treatment, with the argument of saving radiographs. We are very sorry that we have been able to learn about this very important issue in this present series at the time we decided to report about our experience. We hope we will be able to report about final results two years after all the patients in this current series have finished their treatment, and we will intent to analyze at least their final sagittal phenotype, as stated now in the discussion section.
5) Absolutely. No discussion. This population is of high risk for early surgery. The only conclusion we can drawn at this moment is that we have been able to delay early surgery and so we have bought some time. We have also addressed this point in the discussion section. Obviously we cannot talk about efficacy in producing a successful result in the way success is defined in this paper until they finish treatment, reaching maturity, and we will try to report about these results in the future.
6) We are now assessing QoL using validated questionnaires in all our adolescent and adult patients (mainly SRS 22 and ISYQOL among others non scoliosis specific but back pain specific questionnaires), but we do not use it in EOS. At the time these patients were going into puberty we were not using the questionnaires routinely like we are doing now so we expect to be above to report about QoL when we will report about final results when all the patients reaches maturity. This is also addressed now in the discussion section.
7) As we explain now in the material and methods section, we try not to brake the rules. We must say we are very concerned about the potential iatrogenic effect of this type of brace when used improperly or in cases that we consider bad candidates according to the rules of this specific brace design. Thus we do not go under bracing sometimes in patients with moderate curvatures (defined moderate just based on the Cobb angle) and we still can do it (at least technically) in case were the Cobb angle is higher than 60º (the established limit). In all this years we did it only in three cases with a Cobb angle over 60 degrees at presentation, and we did it always in consensus with the orthopedic surgeon and the family after informing them about the non limitation from the technical point of view of using a brace in their cases due to factors like flexibility, low degree of rotation, low torsional index, proper sagittal phenotype and other factor. One of the cases is presented in figure number 4 and this girl showed an initial good response. Being a clear candidate for early surgery, this girl was able to reach a mature stage enough to get final fusion with no need of using growing-rods( mostly due to the behavior of the proximal curve than the main thoracic curve, which is still in the limit of the 50º down, a real grey zone for the decision of surgery, specially in girls with a good esthetic appearance and a good function and balance, like this girl is)
We hope we have clarified all the reviewer's concerns. We thank very much the reviewer for his/her comments and suggestions and for giving us the chance to improve the paper. We are aware about how difficult is for the reviewer giving the green light to articles with such a low level of evidence, but we really made our biggest effort in reporting with the best possible clarity and realistic view about this sensitive issue, which is really lacking of evidence, with the hope that it will help to design better a further study of the highest evidence.
Rebecca Sauvagnac and Manuel Rigo
Round 2
Reviewer 2 Report
I appreciate the authors efforts and modification to this paper. The main strength of this paper is that it includes brace treated early onset cases, as the author have stated, a population seldom targeted in studies concerning conservative treatment. As previously stated, there are some major limitations to the study and I feel that this has been discussed by the authors extensively, in order to clarify drawback to the readers. Considering JCM to be a high-impact journal, there has to be a certain degree of quality to publish papers. I however do feel that the novelty of this paper and interest to the reader is of significance, especially now that major modifications have been applied and drawbacks been discussed.